# Occupational Stress Levels among Radiologists and Radiographers in Hungary during the COVID-19 Era

**DOI:** 10.3390/healthcare12020160

**Published:** 2024-01-10

**Authors:** David Sipos, Orsolya Kövesdi, Bence Raposa, Luca Ferkai, Krisztina Deutsch, Attila Pandur, Árpád Kovács, Melinda Petőné Csima

**Affiliations:** 1Department of Medical Imaging, Faculty of Health Sciences, University of Pécs, Vörösmarty Str. 4, 7621 Pẻcs, Hungary; 2Dr. József Baka Diagnostic, Radiation Oncology, Research and Teaching Center, “Moritz Kaposi” Teaching Hospital, Guba Sándor Str. 40, 7400 Kaposvár, Hungary; 3Doctoral School of Health Sciences, Faculty of Health Science, University of Pẻcs, Vörösmarty Str. 4, 7621 Pẻcs, Hungary; 4Institute of Pedagogy of Health and Nursing Sciences, Faculty of Health Sciences, University of Pécs, Vörösmarty Str. 4, 7621 Pécs, Hungary; 5Department of Oncoradiology, Faculty of Medicine, University of Debrecen, Nagyerdei 98, 4032 Debrecen, Hungary; 6Institute of Education, Hungarian University of Agriculture and Life Sciences, Guba Sándor Str. 40, 7400 Kaposvár, Hungary

**Keywords:** occupational stress, radiographer, radiologist, Effort–Reward Imbalance, ERI, COVID-19

## Abstract

The COVID-19 pandemic has heightened stress levels, potentially affecting the occupational wellbeing of radiographers and radiologists. Our study aimed to assess occupational stress levels within the radiology department and identify contributing factors. A cross-sectional survey was conducted between September and November 2022, with participants comprising radiographers and radiologists affiliated with the Hungarian Society of Radiographers and the Hungarian Society of Radiologists. The online survey collected socio-demographic and COVID-19 data, and the participants completed an effort–reward imbalance questionnaire. The analysis of 406 responses revealed significantly higher effort–reward imbalance (ERI) levels among the radiologists compared to the radiographers (*p* < 0.05). The healthcare professionals with over 30 years of experience exhibited significantly lower ERI levels than those with 1–9 years, 10–19 years, or 20–29 years of experience (*p* < 0.05). Additionally, the individuals aged 31–40 demonstrated higher ERI levels compared to their counterparts aged 19–30, 41–50, and over 51 (*p* < 0.05). The respondents cohabiting with a spouse/partner reported significantly higher stress levels than their single colleagues (*p* < 0.05), while the dog owners exhibited significantly lower ERI levels (*p* < 0.05). Elevated occupational stress highlights specific groups requiring targeted interventions to reduce stress and mitigate burnout among radiologists and radiographers.

## 1. Introduction

Healthcare institution employees are constantly exposed to stress during everyday patient care, which can stem from various sources. These include factors such as long working hours, high workloads, inadequate rewards, exposure to traumatic events, and caring for critically ill patients [1,2,3]. Occupational stress can have adverse effects on the mental and physical health of healthcare workers, causing psychological distress which may lead to anxiety, depression, burnout, or other mental illnesses [1,2,3,4].

The COVID-19 pandemic that occurred at the end of 2019 had a significant impact on the physical and mental health of society as a whole, particularly affecting healthcare workers. Mental health refers to an individual’s emotional, psychological, and social wellbeing, encompassing how a person feels, thinks, manages stress, relates to others, and makes decisions. The pandemic caused a great deal of stress and anxiety as well as triggering feelings of isolation, depression, and grief in people [5,6].

The uncertainty caused by the pandemic brought about significant fear and panic. When fear and anxiety persist, it takes time for the resulting high stress levels to subside. The increasing number of COVID-19 cases, limited resources, and overwhelmed healthcare infrastructure further worsened the inherently stressful situation of healthcare systems [7,8].

It is worth considering that, as a result of all these factors, healthcare workers are more likely to experience burnout and psychological anxiety. Numerous studies have addressed, among other things, the mental health status of healthcare workers during the COVID period, as well as the occupational stress levels of nurses combating coronavirus [1,2,3,6,7].

We must remember the radiographers handling imaging modalities and the radiologist specialists interpreting imaging materials who also work on the front lines. Their work makes the proper diagnosis of patients infected with coronavirus possible [9].

Healthcare professionals employ various coping strategies to manage occupational stress. Many prioritize self-care, incorporating regular exercise, sufficient sleep, and healthy dietary habits to enhance their overall wellbeing. Seeking social support plays a crucial role, as sharing experiences and concerns with colleagues or friends can provide emotional relief [8]. Mindfulness and relaxation techniques, such as meditation or deep-breathing exercises, are commonly used to alleviate stress and promote mental wellbeing. Additionally, time management skills help healthcare professionals organize their tasks efficiently, reducing feelings of being overwhelmed. Lastly, ongoing professional development and training can empower healthcare workers, enhancing their skills and confidence in handling challenging situations, ultimately mitigating stress [10,11].

The presence of animals can significantly contribute to stress coping strategies by providing companionship and emotional support. Interacting with animals, such as pets, has been shown to reduce stress hormones and promote calmness, emphasizing their therapeutic role in enhancing mental wellbeing [12,13,14].

The international literature has reported that radiographers were not recognized as frontline healthcare personnel. Due to the pandemic, radiology departments became overwhelmed, leading to significant psychological stress among the staff [8,15].

Our study sought to evaluate the occupational stress experienced by radiographers and radiologists within a radiology department amid the COVID-19 pandemic, with a specific focus on discerning the underlying factors of said stress. Notably, our investigation included an examination of the impact of pet ownership on the respondents’ occupational stress levels.

## 2. Materials and Methods

Our cross-sectional study was conducted through purposeful, non-random sampling from September 2022 to October 2022. The online survey was distributed to all registered email addresses in the Hungarian Society of Radiographers and the Hungarian Society of Radiologists database. Additionally, we sent the questionnaire to all county hospitals’ radiology departments and shared it through the most popular social media platform (Facebook). The participants were informed about the questionnaire’s content in the first phase of the survey. The questionnaire was completed anonymously, and the participants could interrupt the process at any stage. The respondents confirmed their intention to participate by accepting the content of the consent statement.

During our survey, we included the internationally validated Hungarian version of the effort–reward imbalance questionnaire and a self-designed questionnaire. The latter comprised socio-demographic questions alongside inquiries specific to the COVID-19 virus situation.

In our research, we formulated closed questions regarding socio-demographic characteristics, in line with those found in the international literature, supplemented with questions specific to the pandemic and the occupation of the participants.

Our questions covered the following areas: gender, age, family characteristics (marital status, presence of children, questions related to pets), workplace characteristics (occupation, nature of the current workplace, willingness to take on shifts, weekly working hours, second job).

We formulated questions regarding whether the respondent had ever had a positive COVID-19 test, whether they had been in quarantine, whether they had received training on specific precautions, whether they had examined a patient with COVID-19 during their work, whether there had been a COVID-19 infection in their family, among close friends or colleagues, and their opinion on how the COVID-19 pandemic had influenced societal esteem of healthcare workers, specifically radiographers and radiologists.

The effort–reward imbalance questionnaire consists of three dimensions: efforts at work, workplace rewards, and overcommitment. The abbreviated instrument included fifteen statements, with three about the effort dimension, six about the reward dimension, and an additional six about overcommitment. The statements related to effort and reward specifically assessed stress-inducing factors associated with one’s workplace or job. At the same time, those in the overcommitment category demonstrated individual characteristics appearing in work-related situations for the respondents [16].

The participants completing the questionnaire were required to indicate on a 5-point Likert scale how characteristic each statement was for their work concerning efforts and rewards. In the overcommitment dimension, responses were given on a 4-point Likert scale: (0) Not at all characteristic, (1) Not characteristic, (2) Characteristic, and (3) Very characteristic [16].

The effort scale has a minimum score of 3 and a maximum score of 15. The higher the score on the effort scale, the higher the perceived effort. The reward scale ranges from 6 to 30, where a lower score indicates better rewards. The overcommitment scoring ranges from 6 to 24, with a higher score indicating greater overcommitment by the worker [16].

The ratio of efforts to rewards on the respective scales gives a suitable indicator for measuring occupational stress. This variable, the effort–reward imbalance (ERI), in our case, quantifies the ratio of invested efforts to the rewards received in the daily work of radiographers and radiologists. If ER > 1, the respondent puts in more effort than the rewards received [16].

### 2.1. Statistical Analysis

We conducted descriptive statistics for the sample description (mean, standard deviation) during our research. For comparing the variances in the samples, we applied an F-test. In the case of a normal distribution, we used a two-sample *T*-test, analysis of variance (ANOVA) for group categories, post hoc analysis for within-group differences, and non-parametric distributions. We performed Mann–Whitney U and Kruskal–Wallis tests at a 95% confidence level (*p* < 0.05).

The statistical analysis of the incoming data was carried out using the Statistical Software Package for Social Sciences (SPSS) version 23.0. The graphs and tables were created using Microsoft Excel 2021.

### 2.2. Ethics

We conducted our research with the professional and ethical approval of the Medical Research Council (BMEÜ/253-1/2022/EKU).

## 3. Results

Our research analyzed responses from radiographers and radiologists with work experience exceeding one year, actively involved in patient care during the COVID-19 pandemic. After data cleaning, 406 sets of feedback were included in the analysis.

### 3.1. Socio-Demographic Results

The average age of the sample was 40.9 ± 11.2 (21–67). In terms of gender, the majority of the participants were female (*n* = 324; 79.8%). Those working as radiographers constituted over two-thirds of the total sample (*n* = 287, 70.7%). Regarding age distribution, there was a roughly equal distribution among those aged 19–30 (*n* = 93; 22.9%), 31–40 (*n* = 106; 26.1%), 41–50 (*n* = 97; 23.9%), and those aged 51 or older (*n* = 110; 27.1%). Regarding marital status, over half of the respondents lived with their spouse/partner (*n* = 268; 66.0%), while less than 10% were divorced (*n* = 37; 9.1%), and fewer than a quarter were single (*n* = 87; 21.4%). Almost half of the respondents had not had children (*n* = 164; 40.4%) up to their completion of the questionnaire. (Table 1).

Regarding the nature of the main job of the radiologists and radiographers, most worked in state healthcare (*n* = 368; 90.6%). Slightly over 30% of the respondents had a second job (*n* = 129, 31.8%). The sample predominantly comprised healthcare workers with 1–9 years of experience (*n* = 159; 39.2%). Following this group were those with over 30 years of professional experience (*n* = 115; 28.3%). Those with 10–19 years (*n* = 72, 17.7%) and 20–29 years (*n* = 60; 14.8%) of experience in healthcare showed a similar distribution. Most respondents worked 40 h per week (*n* = 313; 77.1%). Two-thirds of the sample (*n* = 271; 66.7%) participated in on-call duties during their work, with those taking on more than three on-call shifts per month being the majority (*n* = 217; 53.4%). (Table 1).

The number of pet owners (*n* = 217; 53.4%) and those without pets (*n* = 189; 46.6%) was almost evenly distributed among the respondents. Ninety individuals (22.2%) had only a dog, and forty-four individuals (10.8%) owned both a dog and a cat. Most of the dog owners (*n* = 44, 32.83%) took their dogs for a walk once a day (Table 1).

### 3.2. Sample Characteristics Related to COVID-19

Most of the respondents (*n* = 370; 91.1%) had examined a patient with COVID-19 during their work, but only slightly more than half of the sample reported having had a coronavirus infection (*n* = 226; 55.7%). Most of them (*n* = 342; 84.2%) mentioned that there had been cases of coronavirus illness among their family, friends, and colleagues as well (Table 2).

We asked questions regarding the training healthcare workers received for adequate protection against coronavirus. The majority read informational materials (*n* = 145; 35.7%), but many also prepared through e-learning (*n* = 91; 22.4%) or workplace simulations (*n* = 72; 17.7%) for the fight against coronavirus. Some reported receiving no specific highlighted training for the virus (*n* = 89; 22%). (Table 2).

Regarding societal esteem of radiographers and radiologists, in both cases, more than half of the sample (*n* = 231, 56.9%; *n* = 222, 54.7%) believed that it had not changed due to the pandemic.

### 3.3. The Results of the Effort–Reward Imbalance Questionnaire

The descriptive statistics of the effort–reward imbalance questionnaire, the reliability values indicated using Cronbach’s alpha for its scales, and the overall stress score for the entire sample are summarized in Table 3.

### 3.4. The Relationship between Socio-Demographic Characteristics and Occupational Stress

The occupational stress score of the radiologists (0.87 ± 0.41) was significantly higher than that of the radiographers (0.49 ± 0.23) (*p* < 0.001).

The respondents with a side job had a significantly higher stress score (0.70 ± 0.39) than their counterparts working only their main job (0.56 ± 0.30) (*p* < 0.001).

The occupational stress score of those with more than 30 years of experience in the profession (0.50 ± 0.24) was significantly lower than that of the colleagues with 1–9 years (0.61 ± 0.30), 10–19 years (0.70 ± 0.41), and 20–29 years (0.65 ± 0.44) of experience (*p* = 0.006; *p* = 0.001; *p* = 0.007).

The stress score of the individuals living alone (0.48 ± 0.22) was significantly lower than that of the respondents who were divorced (0.67 ± 0.32) or living with a spouse or partner (0.64 ± 0.37) (*p* = 0.004; *p* = 0.001).

Pet ownership alone was not a significant factor in the occupational stress scores (*p* = 0.374). However, analyzing groups of respondents with pets revealed a significant relationship between pet type and stress score changes (*p* = 0.006). The respondents with only cats (0.72 ± 0.39) had significantly higher stress scores than those with dogs (0.51 ± 0.22). Furthermore, the respondents without pets (0.62 ± 0.34) had significantly higher stress scores than the dog owners (*p* = 0.012) but significantly lower stress scores than the cat owners (*p* = 0.042).

(All the results can be seen in Table 1).

### 3.5. The Relationship between COVID-19 Characteristics and Occupational Stress

The respondents’ occupational stress scores were not significantly influenced by whether the respondent had already had a COVID-19 infection (*p* = 0.826), and the examination of a COVID-19-infected patient also did not have a significant impact (*p* = 0.484).

However, the occupational stress scores of the respondents were significantly higher for those who had experienced COVID-19 infections within their family, among close friends, and colleagues (0.63 ± 0.35) compared to those for whom COVID-19 had occurred only among colleagues (0.47 ± 0.22) (*p* = 0.04). (All the results can be seen in Table 2).

## 4. Discussion

Elevated occupational stress is a common phenomenon in various professions, which can significantly impact the physical and mental health of employees and their workplace performance [1,2,3,4,5,6,7,8]. During public health emergencies, such as the COVID-19 pandemic, healthcare workers are exposed to even higher stress levels, exacerbated by increased risk of infection, heavier workloads, and a lack of social support [1,2,3,4,6,7].

Birgit et al. also highlighted the role of one’s occupation in ERI, where healthcare professionals typically experienced a more significant imbalance in their effort–reward compared to physicians [17].

Our study results confirmed that the nature of one’s occupation is a significant factor in occupational stress; however, in our case, the radiologists showed a higher effort–reward imbalance than the radiographers. Our survey aligns with the findings of the study conducted by Nguyen Van et al. [18], who found that in some cases there is more perceived reward than perceived effort in the observed workplace.

For years, professionals researching this field believed financial rewards to be the most effective way to motivate employees and boost morale. There are other ways to reward employees beyond financial compensation. Some of these include recognition and praise from their superiors, providing opportunities for their development, and taking on meaningful projects [19,20].

Msaouel et al. [19] demonstrated that the 31–40 age group had a significantly higher effort–reward imbalance than their younger or older colleagues. This is consistent with our study results, as we also found that the age group between 31 and 40 years of age exhibits a significantly higher stress level than the younger and older age groups.

Based on Nguyen Van et al.’s research [18], female healthcare workers are more likely to report a lower ER ratio. The survey identified higher effort and overcommitment among men, but they received lower rewards than women. Our survey does not exhibit this trend. In terms of gender, no significant difference was observed in the dimensions of effort, rewards, or overcommitment.

Yubonpunt and colleagues [21] found that employees living with their partners showed higher coping scores, indicating that they handled stress more effectively than their single counterparts. This could be explained by viewing their partners as a secure zone providing emotional support. Their research revealed that healthcare workers with children had lower stress levels and higher coping scores compared to their childless counterparts.

Ali et al. [22] investigated the main stressors and coping strategies of frontline nursing staff during the COVID-19 pandemic in Alabama. Their results showed that single nurses and those without children had significantly higher stress levels than their counterparts in relationships.

However, some studies have found the opposite and identified living with a spouse or partner and having children as a source of concern and stress [22,23].

Among respondents with children, higher stress levels may have been observed because, during quarantine, online education could impose a significant burden on parents in terms of its undertaking and the placement of children. This could be particularly true for healthcare workers, given the nature of their work, which often did not allow for remote work. In cases where both parents were healthcare professionals, the arrangement and supervision of young children may have been even more complicated.

Hossain et al. [23] examined the relationship between COVID-19 and stress based on marital status. It was found that the stress levels of doctors and healthcare workers were significantly higher than those of teachers, engineers, or researchers. Furthermore, the highest stress levels and severe anxiety were measurable among married individuals, while singles exhibited mild stress levels and moderate anxiety [23].

Our present study found that the stress levels of single respondents were significantly lower than those living with a spouse or partner. Additionally, there was no significant connection between the respondents’ stress levels and whether they had children, indicating that having children was not interpreted as a protective factor or a significant stressor. These results align with Hossain’s findings [23], and our study may support the assumption that individuals in a marital relationship may have had increased concerns regarding their daily livelihood and protecting their family members from infection.

According to our survey, the number of years spent in healthcare showed a significant relationship with occupational stress. The respondents working in healthcare for over 30 years exhibited the lowest stress levels in the examined sample. This might be attributed to the wealth of work experience accumulated over the years, which could have assisted them in coping with the challenges posed by the pandemic.

Mainly during the first wave, radiographers in England reported on protocols that changed daily, sometimes hourly, which was one of the causes of the anxiety felt by many. The rapidly changing information primarily focused on the use of personal protective equipment [24].

Foley et al. [15] also surveyed the training received by radiographers about the pandemic, reporting similar results to our study. In their case, the most common form of education was practical training, received by 28% of the workers. This was followed by nearly 20% of respondents who read publications or materials related to COVID-19. It is evident that various forms of training were employed in both Hungary and Ireland to prepare and inform radiographers about combating COVID-19.

Pets are an integral part of the daily lives of pet owners. During the COVID-19 pandemic, the number of household pets exponentially increased worldwide, although the reasons for this growth are not yet fully understood [25]. We observed that the respondents without pets had a significantly higher stress level than the individuals with dogs; however, their stress level was significantly lower than that of cat owners. Additionally, those who only owned cats reported significantly higher stress than those who owned dogs. Based on our results, pet ownership alone does not constitute an apparent predictive factor for the stress levels of healthcare workers. However, the type of pet may be an influencing factor. Understanding the impact of pets on human mental health and wellbeing, especially during the pandemic, when we lived isolated lives for an extended period of time, is crucial for determining whether animals can be incorporated into preventive or restorative interventions to promote mental health and reduce stress levels.

The aim of Grajfoner et al.’s [25] research was to explore how pets affected the mental health of their owners during the COVID-19 pandemic. Their results showed that the respondents with pets enjoyed significantly better mental health than those without animals. This is because pet owners felt that they coped better with adverse situations and experienced much more positive emotions during the quarantine.

Exercise is an essential tool in treating and preventing various physical illnesses. Additionally, it has long been recognized that regular physical activity comes with benefits in alleviating depressive and anxious symptoms [26,27].

The media plays a role in maintaining connections among people and promoting emotional stability, which, as previous studies have pointed out, is crucial because family support plays a vital role in the stress management of healthcare workers [28]. Additionally, numerous pages and groups were formed on social platforms where videos, various materials on mental health, or relaxation exercises were shared. All of this could have assisted those going through depressive phases due to various aspects of the pandemic. The critical role of the media lies in providing people with credible information and striving to keep them connected during these challenging times [28,29].

Evanoff et al. [30] emphasized that organizational and societal support, clear communication, and a sense of control were protective factors for mental health during the pandemic.

Training programs that enhance employees’ resilience can improve burnout rates and other wellbeing outcomes, but organizational-level interventions that reduce perceived job demands or increase resources are generally more effective [4,5,6,8].

Windarwati et al. [31] found that 97.9% of respondents evaluated that their workplace rewarded them for their work. The most significant stress-inducing factor for them was the daily use of personal protective equipment. The three most commonly used coping strategies by healthcare workers were adopting positive behavioral patterns to maintain self-motivation, reading information about COVID-19 prevention and the prevention of its spread, and following and adhering to appropriate and recommended self-defense tools and measures. Additionally, their study highlighted family support as a significant factor in the motivation of these healthcare workers in the battle against the pandemic, as seen in several previous surveys.

Our study did not explore the stressors related to individual protective equipment or address issues concerning the availability of protective equipment. However, these can be a noteworthy aspect of factors influencing occupational stress, as healthcare workers must use individual protective equipment daily to protect themselves, each other, and their patients. Using such equipment could cause discomfort, such as unwanted skin reactions, difficulty in breathing, heat stress, dizziness, and nausea.

Lewis et al. [32] surveyed South African radiographers’ experiences of COVID-19. Their results also confirm that radiographers reported feeling sad, fearful, confused, stressed, frightened, anxious, and overwhelmed. The constant fluctuation of these emotions, akin to a rollercoaster, took a toll on them mentally and physically. The mental health of radiographers was further compromised by the inability to see their families and friends, causing anxiety about the risk of infecting them.

We need to mention that machine learning in radiology can alleviate the burden on radiologists and radiographers by automating routine tasks, allowing them to focus on more complex analyses and patient interactions. This can potentially reduce occupational stress by streamlining workflows, enhancing efficiency, and improving overall job satisfaction in the medical imaging field [33,34].

## 5. Conclusions

The critical role of radiographers and radiologists in service delivery is not widely understood among healthcare workers. Therefore, it is crucial to emphasize a more significant promotion of the radiography profession.

Based on our findings, we discovered that factors such as profession, family status, presence of a side job, number of years spent in healthcare, type of household pet, and occurrence of coronavirus infection can significantly affect the levels of occupational stress experienced by radiologists and radiographers. With these findings in mind, targeted protective interventions can be implemented for this group of affiliated health professionals in order to minimize the chronic effects of workplace stress on individuals.

## 6. Limitations

As a limitation of this research, our results are indicative, considering that our survey was based on cross-sectional sampling, thus reflecting a specific condition. Using exclusively close-ended questions is also a limitation of our study. Another limitation could lie in the time constraints and stressful situations arising from the burden caused by the pandemic, which may have negatively influenced health professionals’ willingness to respond.

## Figures and Tables

**Table 1 healthcare-12-00160-t001:** Relationship between socio-demographic features and occupational stress.

Variable		*n* (%)	ERI	*p*-Value
Gender	Male	82 (20.2)	0.63 ± 0.35	*p* = 0.432
	Female	324 (79.8)	0.60 ± 0.34	
Profession	Radiographer	287 (70.7)	0.49 ± 0.23	*p* < 0.001
	Radiologist	119 (29.3)	0.87 ± 0.41	
Age (years)	19–30	93 (22.9)	0.57 ± 0.28	*p* = 0.062
	31–40	106 (26.1)	0.67 ± 0.34	
	41–50	97 (23.9)	0.57 ± 0.28	
	51+	110 (27.1)	0.60 ± 0.42	
Family status	Living with spouse	268 (66.0)	0.64 ± 0.37	*p* < 0.001
	Divorced	37 (9.1)	0.67 ± 0.32	
	Single	87 (21.4)	0.48 ± 0.22	
	Other	14 (3.4)	0.53 ± 0.21	
Do you have a child?	One	97 (23.9)	0.63 ± 0.38	*p* = 0.608
	More than one	145 (35.7)	0.59 ± 0.34	
	None	164 (40.4)	0.60 ± 0.31	
Workplace	National	368 (90.6)	0.60 ± 0.32	*p* = 0.876
	Private	38 (9.4)	0.68 ± 0.46	
Side job?	Yes	129 (31.8)	0.70 ± 0.39	*p* < 0.001
	No	277 (68.2)	0.56 ± 0.30	
Years spent in healthcare	1–9	159 (39.2)	0.61 ± 0.30	*p* = 0.006
	10–19	72 (17.7)	0.70 ± 0.41	
	20–29	60 (14.8)	0.65 ± 0.44	
	30+	115 (28.3)	0.50 ± 0.24	
Hours worked per week	Less than 40 h	67 (16.5)	0.61 ± 0.31	*p* = 0.868
	40 h	313 (77.1)	0.59 ± 0.34	
	More than 40 h	26 (6.4)	0.63 ± 0.39	
On-call shift	None	135 (33.3)	0.57 ± 0.34	*p* = 0.15
	1–2	54 (13.3)	0.68 ± 0.31	
	3+	217 (53.4)	0.60 ± 0.34	
Do you have a pet?	Yes	217 (53.4)	0.59 ± 0.34	*p* = 0.374
	No	189 (46.6)	0.62 ± 0.34	
What kind of pet	Dog	90 (22.2)	0.51 ± 0.22	*p* = 0.002
	Cat	55 (13.5)	0.72 ± 0.39	
	Dog and cat, as well	44 (10.8)	0.60 ± 0.40	
	Other	28 (6.9)	0.57 ± 0.37	
	None	189 (46.6)	0.62 ± 0.34	

**Table 2 healthcare-12-00160-t002:** Relationship between COVID-19-related variables and occupational stress.

Variable		*n* (%)	ERI	*p*-Value
Have you examined COVID-19 infected patient?	Yes	370 (91.1)	0.60 ± 0.34	*p* = 0.484
	No	36 (8.9)	0.64 ± 0.32	
Has COVID-19 infected you?	Yes	226 (55.7)	0.62 ± 0.39	*p* = 0.826
	No	180 (44.3)	0.58 ± 0.26	
What COVID-19 precautions you have taken?	Did not receive special training	89 (22)	0.57 ± 0.33	*p* = 0.529
	Informational articles	145 (35.7)	0.68 ± 0.41	
	Workplace simulation	72 (17.7)	0.56 ± 0.28	
	E-learning	91 (22.4)	0.56 ± 0.26	
	Other	9 (2.2)	0.46 ± 0.13	
COVID-19 occurred:	Among colleagues	47 (11.6)	0.47 ± 0.22	*p* = 0.012
	Among family, close friends and colleagues	342 (84.2)	0.63 ± 0.35	
	Among close friends and family	10 (2.5)	0.58 ± 0.21	
	Did not happened	7 (1.7)	0.43 ± 0.13	

**Table 3 healthcare-12-00160-t003:** The respondents’ values for the ERI questionnaire dimensions, Cronbach’s alpha values, and stress scores.

	Number of Statements	Mean	SD	Min	Max	Cronbach-Alpha	
Effort	3	7.78	3.40	3	15	0.77	ERI 0.60 ± 0.34
Reward	6	14.55	5.89	6	30	0.80	
Overcommitment	6	14.73	3.14	6	24	0.57	

## Data Availability

Data are contained within the article.

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
