# Peer review of "Occupational Stress Levels among Radiologists and Radiographers in Hungary during the COVID-19 Era"

_healthcare, 2024, doi:10.3390/healthcare12020160_

Round 1
Reviewer 1 Report
Comments and Suggestions for Authors
The motivation of the study presented is not clear. The novelty and contributions should clearly be presented in the introduction. What are the potential applications of this study? The number of paragraphs throughout the manuscript should be reduced.
Comments on the Quality of English LanguageSeveral grammatical errors throughout the manuscript. They should be corrected.
Author Response
Dear Reviewer 1!
Thank you sincerely for your thoughtful and constructive feedback on our work. Your insights have been invaluable in refining and improving the quality of our research. We appreciate your time and expertise.
Regarding your comments:
The motivation of the study presented is not clear. The novelty and contributions should clearly be presented in the introduction. - Thank you for your observation. In the introduction, we emphasize multiple times that radiographers and radiologists are not considered among the front-line workers, despite their clear roles in capturing diagnostic images and analyzing them. The communication focuses on the workplace stress affecting them and the underlying reasons behind it. We have rephrased the relevant part of the introduction.
What are the potential applications of this study? Given our results, profession, family status, the presence of side job, the number of years spent in healthcare, the type of household pet, and the occurrence of coronavirus infection can significantly influence the occupational stress levels of radiologists and radiographers. With awareness of these findings, targeted protective interventions can be implemented for the examined group of allied health professionals to minimize the chronic effects of workplace stress on the individual. Corrected at the Conclusions section.
The number of paragraphs throughout the manuscript should be reduced. – We tried our best, thank you once again for your comments.
Reviewer 2 Report
Comments and Suggestions for Authors
Rewrite the abstract based on the strobe checklist, https://www.strobe-statement.org/checklists/ conference abstracts.
Line 25, what’s ERI??
This doi: 10.1007/s12029-021-00752-5. And doi: 10.1002/npr2.12179 Can be useful for introduction
In introduction, gap and purpose should be expressed in better sentence.
Our cross-sectional, descriptive, this isn’t descriptive, in result p value an analytic statistics presented.
purposeful, non-random sampling. Because of this issue, generalizability is difficult, it should be stated in limitations
Rewrite the method based on the strobe checklist, the sequence of the contents is not appropriate. Study design. Setting, Participants, sample size calculation, Variables, Data sources/ measurement, must be clear and complete.
Line 196-245 is very boring and unscientific and it is better to express it in the table.
In the discussion, all the sentences need a reference. Also it is better to be more concise.
Limitations and Conclusions should be more concise.
Author Response
Dear Reviewer 2!
We extend our sincere gratitude for your thorough review of our work. Your constructive feedback and valuable suggestions have significantly enhanced the quality of our research. We appreciate your time, expertise, and thoughtful contributions to the refinement of our manuscript.
Regarding your comments:
Rewrite the abstract based on the strobe checklist, https://www.strobe-statement.org/checklists/ conference abstracts. – Thank you for your comment, we rewrote it
Line 25, what’s ERI?? – ERI value is the ratio of Effort and Reward
This doi: 10.1007/s12029-021-00752-5. And doi: 10.1002/npr2.12179 Can be useful for introduction – Thank you for raising our attention – the mentioned articles have been used
In introduction, gap and purpose should be expressed in better sentence. – Carried out
Our cross-sectional, descriptive, this isn’t descriptive, in result p value an analytic statistics presented. purposeful, non-random sampling. Because of this issue, generalizability is difficult, it should be stated in limitations. – Thank you for your comment, we made corrections.
Rewrite the method based on the strobe checklist, the sequence of the contents is not appropriate. Study design. Setting, Participants, sample size calculation, Variables, Data sources/ measurement, must be clear and complete. – Thank you – corrections have been made.
Line 196-245 is very boring and unscientific and it is better to express it in the table. – We tried to rewrite it. The results were listed in the Table 1.
In the discussion, all the sentences need a reference. Also it is better to be more concise. – We rewrote the discussion section, also inserted many references.
Limitations and Conclusions should be more concise –Thank you, corrections have been made.
Reviewer 3 Report
Comments and Suggestions for Authors
This paper is interesting, but it needs some reworking. See comments in the attached report.

Author Response
Dear Reviewer 3,
Thank you immensely for your insightful and detailed review of our manuscript. Your thoughtful comments and suggestions have proven invaluable in strengthening the overall quality of our work. We are genuinely appreciative of your time and expertise in contributing to the refinement of our research.
Regarding your comments:
Quite a large number of articles have been written on stress in healthcare workers
during the COVID-19 pandemic. Here is one more.
It is neither a bad nor an outstanding paper. It is interesting, but it needs to be
improved. The format is adequate, the quality of language is correct, but some points
need to be improved, for instance:
- In the abstract, the authors write: “Those working in healthcare for more than 30
years had significantly lower 24 ERI levels than those who had been in the
profession for 1-9 years, 10-19 years, or 20-29 years 25 (p<0.05).” However, in
Table 1 it is those with 20-29 years’ experience in Healthcare who have the
lowest ERI, then those with 10-19 years. – I’m really sorry but I think you are not right. The values in table and abstract are correct.
- Long segments of the discussion should in fact appear in the introduction; for
example, a survey of South African radiographers appears in the last paragraph
before the conclusion. – Thank you, corrections have been made.
- The review of the literature is a bit short; the references are relevant, but so
much has been written on the subject that one would expect a more thorough
search. In fact, as mentioned above, a substantial part appears only in the
discussion section. – Thank you, corrections have been made.
- Having written that “Healthcare professionals employ various coping strategies
to manage occupational stress” and proceeded to mention quite a few self-care
strategies, the authors then focus uniquely on possession of a pet: why? – Thank you for your comment. Many of the co-authors have pet at home. As we discussed at the beginning of our research, it was quite good to have a “support by them”. Also we started to read literature about the pets and their effect on mental health.
This paper could be interesting, but it need to be reworked; it cannot be published as it stands
Round 2
Reviewer 1 Report
Comments and Suggestions for Authors
Nowadays Machine Learning is quite into medical imaging and can assist radiologists well. This point should be included and discussed.
Comments on the Quality of English LanguageSome minor proof reading is required.
Author Response
Dear Reviewer 1!
Once again thank you sincerely for your thoughtful and constructive feedback on our work. Your insights have been invaluable in refining and improving the quality of our research. We appreciate your time and expertise.
Regarding your round 2 comments:
Nowadays Machine Learning is quite into medical imaging and can assist radiologists well. This point should be included and discussed. – THANK YOU, A PARAGRAPH HAS BEEN INSERTED
Reviewer 2 Report
Comments and Suggestions for Authors
There is no more comments.
Author Response
Dear Reviewer 2!
We extend our sincere gratitude for your thorough review of our work. Your constructive feedback and valuable suggestions have significantly enhanced the quality of our research. Once again we appreciate your time, expertise, and thoughtful contributions to the refinement of our manuscript.
Regarding your round 2 comments:
There is no more comments. – THANK YOU VERY MUCH
Reviewer 3 Report
Comments and Suggestions for Authors
Definitely a much improved version worth publishing. Most of my previous comments have been taken into consideration.
Comments on the Quality of English LanguageJust a final check, but no major concern about the quality of language.
Author Response
Dear Reviewer 3,
Once again thank you immensely for your insightful and detailed review of our manuscript. Your thoughtful comments and suggestions have proven invaluable in strengthening the overall quality of our work. We are genuinely appreciative of your time and expertise in contributing to the refinement of our research.
Regarding your round 2 comments:
Definitely a much improved version worth publishing. Most of my previous comments have been taken into consideration. – WE APPRECIATE YOUR FEEDBACK. THANK YOU VERY MUCH